# Examining the acceptability of actigraphic devices in children using qualitative and quantitative approaches: protocol for a systematic review and meta-analysis

Anna Charlotte Morris [iD],[1] Laurence Telesia,[1] Alice Wickersham [iD],[1]
Sophie Epstein [iD],[1] Faith Matcham,[2] Edmund Sonuga-Barke,[3] Johnny Downs [iD][1]

[1]CAMHS Digital Lab, Dept of Child and Adolescent Psychiatry, Institute of Psychiatry Psychology and Neuroscience, King's College London and South London and Maudsley NHS Foundation Trust, UK, London, UK
[2]School of Psychology, University of Sussex, Brighton, UK
[3]Dept of Child and Adolescent Psychiatry, Institute of Psychiatry, Psychology and Neuroscience, King's College London, London, UK

**Correspondence to**
Anna Charlotte Morris;
anna.morris@kcl.ac.uk

## ABSTRACT

**Introduction** Actigraphy is commonly used to record free living physical activity in both typically and atypically developing children. While the accuracy and reliability of actigraphy have been explored extensively, research regarding young people's opinion towards these devices is scarce. This review aims to identify and synthesise evidence relating to the acceptability of actigraphic devices in 5–11 year olds.

**Methods and analysis** Database searches will be applied to Embase, MEDLINE, PsychInfo and Social Policy and Practice through the OVID interface; and Education Resources Information Center (ERIC), British Education Index and CINAHL through the EBSCO interface from January 2018 until February 2023. Supplementary forward and backward citation and grey literature database searches, including Healthcare Management Information Consortium (HMIC) and PsycEXTRA will be conducted. Qualitative and quantitative studies, excluding review articles and meta-analyses, will be eligible, without date restrictions. Article screening and data extraction will be undertaken by two review authors and disagreements will be deferred to a third reviewer. The primary outcome, actigraphic acceptability, will derive from the narrative synthesis of the main themes identified from included qualitative literature and pooled descriptive statistics relating to acceptability identified from quantitative literature. Subgroup analyses will determine if acceptability changes as a function of the key participant and actigraphic device factors.

**Ethics and dissemination** Ethical approval is not required for this systematic review as it uses data from previously published literature. The results will be presented in a manuscript and published in a peer review journal and will be considered alongside a separate stream of codesign research to inform the development of a novel child-worn actigraphic device.

**PROSPERO registration number** CRD42021232466.

## STRENGTHS AND LIMITATIONS OF THIS STUDY

⇒ Detailed study methodology and statistical analysis strategy are planned, to enable the inclusion of quantitative and qualitative literature.
⇒ Comprehensive search terms are informed by a theoretical framework to enhance the likelihood of detecting literature relevant to acceptability.
⇒ Literature not written in English will be excluded and may lead pertinent studies to being omitted.
⇒ Conclusions that can be surmised from the evidence of this systematic review could be restricted by limited actigraph monitoring protocol adherence and attrition data reported in quantitative studies and the potentially low quality of identified literature.
⇒ Narrative synthesis of qualitative evidence may be influenced by reviewer subjectivity and must be critically appraised.

## BACKGROUND

Actigraphy, an objective measure of movement, is widely considered the preferred method for activity-based monitoring in clinical and epidemiological research.[1] Modern actigraphic devices, known as actigraphs are small, inexpensive and readily accessible rendering them ideal for inclusion in studies investigating free living patterns of physical activity and sleep-wake cycles.[2] Actigraphs are credited for their ability to overcome important measurement biases associated with traditional subjective methods including memory, perception and social desirability bias to produce a more detailed and accurate account of motor activity.[3] These benefits are particularly pronounced in the study of young children, where parent or teacher-reported questionnaires are often relied on as the primary source of information[4] and where self-reported measures tend to be inaccurate.[5 6]

Actigraphs have been applied extensively in the remit of children's physical and mental health, especially in the context of sleep medicine, where actigraphic devices are used

to record nocturnal movement patterns in children.[7] Actigraphic devices are increasingly relied on to examine sleep patterns in childhood neurological disorders[8] and neurodevelopmental disorders.[9][10] Deficiencies in sleep quantity or quality are also linked to the onset of various health difficulties,[11][12] behavioural problems and cognitive impairments.[13] Therefore, objective measures of sleep are increasingly used to support clinical assessment and management of sleep disturbances or disorders.[4] Compared with more costly data collection methods such as polysomnography which typically records sleep information for one or two nights in a laboratory, actigraphy can be used in naturalistic environments for extended periods,[14] making it easier to collect data from children who might find it difficult to sleep normally in unfamiliar surroundings.[15][16]

Considerable efforts have also been made to deploy actigraphs to assess physical activity in children, including obesity,[17][18] and chronic conditions such as cerebral palsy.[19] Particular attention is also being paid to the potential role of actigraphy in the assessment and diagnosis of attention deficit hyperactivity disorder (ADHD),[20] a common childhood disorder characterised by the persistent presence of three core symptoms, namely, impulsivity, inattention and hyperactivity.[21][22] Within clinical practice, diagnosis of ADHD is usually based on a clinical assessment of the individual, supplemented by collateral history and reports from parents/carers and teachers.[21][23–25] However, the clinical utility of these measures has been questioned, as the main features of the disorder are rarely observed directly or systematically measured by members of the clinical decision-making team.[26] In contrast, actigraphy provides the means to objectively monitor physical activity, a well-established means of determining the severity of hyperactive symptoms,[27] across a variety of different contexts.

There is a considerable body of evidence focusing on the accuracy of actigraphic outcomes in the context of physical activity and sleep;[28–30] however, the question of acceptability has often been overlooked. Engagement with medical devices is heavily dependent on aesthetic preferences[31] and is potentially even more pertinent for children, who may have unique physical feature requirements compared with adults.[32] Children with neurodevelopmental disorders may also experience particular tactile sensitivities associated with the actigraphic device or its placement,[33] and stigma associated with visible markers of medical investigation.[34] Therefore, it is essential to design actigraphs that incorporate the needs and preferences of children to ensure sustained engagement with these products, particularly when asking children to wear them continuously over prolonged periods in-home and school-based environments.

## Aims

This review aims to examine the literature pertaining to the acceptability of body-worn actigraphic devices for children aged 5–11 years old. While this review may discover overlap between features considered acceptable by both typically and atypically developing children, differences between these groups will be highlighted if identified. If available, we aim also to include feedback on acceptability from parents/caregivers of young children who may provide insightful feedback about the acceptability of these devices. As such, the results of this review may have several practical applications, both in typically and atypically developing children. For example, by offering information to clinical teams on the types of actigraphic devices that are likely to be well tolerated by children under their care or guiding the development of future study protocols to include child-accepted actigraphs.

Furthermore, we aim to use the findings from this review alongside data gathered from concurrent codesign research, to inform the design of a bespoke actigraphic device primarily indented to monitor physical activity within this age group.

## METHODS AND ANALYSIS

This protocol has been prospectively recorded in the PROSPERO international database. We used the Preferred Reporting Items for Systematic Reviews and Meta-Analyses protocols (online supplemental file 1) checklist when writing our report.[35]

## Eligibility criteria
### Population
We are principally interested in actigraphic acceptability among primary school-aged children. In the UK, this represents children aged 5–11, an age range that typically encompasses most primary school-aged children worldwide. Therefore, studies will be included in this review if they involve children (where at least 50% fall within the 5–11 year age range or where the average age falls within this age range) in the design, development or evaluation of an actigraphic device. Additionally, parents and caregivers of young children often act as the 'gatekeeper' responsible for enabling or restricting access to or time spent interacting with wearable technologies;[36][37] therefore, studies including parents or caregiver/s of children aged between 5 and 11 years old will also be included.

### Exposure and outcome
Studies will be eligible for this review if they investigate the development or use of an actigraph and report a corresponding measure of acceptability. In this review, acceptability will be defined according to the theoretical framework of acceptability (TFA),[38] including participants' affective (eg, feelings), cognitive (eg, perceptions) or behavioural (eg, attrition rates) response towards the actigraph. Included studies may use actigraphs for any purpose, as the primary focus of the study or as part of a multifaceted tool. Participants must be required to wear an actigraphic device in a community setting to be included in this review. Studies will be excluded if they required the wearable device to be worn in only a

**Table 1** Inclusion/exclusion table for studies exploring actigraphic devices in children

|  | Inclusion criteria | Exclusion criteria |
|---|---|---|
| Publication status | Studies written in English<br>Published or unpublished studies | Studies written in other languages<br>Commentaries, opinion pieces, editorials, magazine articles and technical or internal reports |
| Study design | Qualitative or quantitative studies | Reviews or meta-analyses |
| Device | Studies in which a wearable activity tracker to measure motor activity was worn by a child within the target age group | Studies that do not employ any wearable devices<br>Studies using wearable devices which measure parameters other than motor activity (eg, balance, number of steps, proportion of time sitting etc.)<br>Studies in which wearable activity trackers were discussed, but not worn |
| Age | Studies where all participants are aged between 5 and 11 years, as indicated by age range, or reported school year<br>Studies reporting a measure of central tendency (eg, median, mean) which falls within the range of 5–11 years<br>Studies that present data in a way that allows extraction for a subgroup aged between 5 and 11 years<br>Studies that satisfy the above age criteria for the child wearing the device, but feedback is provided by a caregiver or other relevant stakeholder | Studies where all participants are outside the target age of 5–11 years<br>Studies where it is not possible to extract data specific for participants aged between 5 and 11 years |
| Setting | Studies with protocols which involves the wear of an activity tracker in a community setting, regardless of whether the device is also worn in a non-community setting | Studies that take place entirely in a non-community setting (eg, laboratory, university or hospital setting) |
| Acceptability | Studies which present a qualitative or quantitative measure of acceptability<br>Measures can include focus group or interview feedback, proportions of participants declining to wear the device or proportion and extent of deviation from the study's wear protocol | Studies that do not contain any direct or indirect measure of acceptability |

non-community setting, for example, lab-based, university or hospital setting.

## Study design

Qualitative studies, specifically codesign, focus groups and interview methodologies will be included as well as quantitative studies including feasibility, pilot, observational cohort or intervention studies. Reviews, meta-analyses and protocols will be excluded, as will commentaries, opinion pieces, editorials, magazine articles and technical or internal reports. As patient-centred design and acceptability literature is an emerging field, we will extend inclusion to published conference abstracts if they provide sufficient information to demonstrate that they meet the inclusion criteria. Likewise, letters that describe the methods and results of otherwise unpublished research may also be considered for inclusion. Searches will be conducted from January 2018 until February 2023 as this timeframe will capture the recent rapid expansion of wearable technology but is also likely to include publications which use older devices that are still suitable for use and potentially pertinent to current-day children. Studies can be published or unpublished but must be written

in English. See table 1 for more detailed inclusion and exclusion criteria.

## Information sources and search strategy

The electronic databases Embase, MEDLINE, PsychInfo and Social Policy and Practice through the OVID interface; and Education Resources Information Center (ERIC), British Education Index and CINAHL through the EBSCO interface will be searched using a combination of free text and thesaurus terms.

The search terms will derive from three key concepts, children (population), actigraphy (exposure) and acceptability (outcome). Search terms and advanced search techniques will be used, including truncation, wild card and adjacency operators, which will be tailored for each database as required. We will also review the reference list of included articles and related existing review articles (backward citation searching), and articles which have referenced included studies (forward citation searching). Both forward and backward citation searching will be conducted in Web of Science Core Collection, and if a citation is unavailable in this source, Google Scholar will be queried.

Additional grey literature searches will be conducted within PsycEXTRA, a complementary database to PsychINFO and the Healthcare Management Information Consortium (HMIC), comprising records from the Department of Health (DoH) and the King's Fund. Supplementary searches will be carried out using the same search strategy described above for the primary electronic databases. In addition, consultation with experts will be conducted and authors of studies included in the review will be contacted to inquire about any known studies matching the inclusion criteria. Full search strategies for the database are provided as follows; Embase (online supplemental file 2), MEDLINE (online supplemental file 3), PsychInfo (online supplemental file 4), Social Policy and Practice (online supplemental file 5), ERIC (online supplemental file 6), British Education Index (online supplemental file 7) and CINAHL (online supplemental file 8).

### Data management

The search output will be downloaded into EndNote V.20 reference management software. Duplicates will automatically be detected and then removed by the reviewers. The remaining references will be exported to a Microsoft Excel document and manually screened for undetected duplicates.

### Study selection process

Two reviewers (ACM and LT) will each independently screen a 10% sample of the titles and abstracts generated by the search strategy. Interrater reliability for this sample will be determined using Cohen's kappa.[39] If the interrater reliability is considered 'substantial' or 'almost perfect', based on a kappa statistic greater than 0.60[40] each review author will independently review 50% of the remaining abstracts, with the option to jointly discuss unclear or otherwise challenging abstracts. If the strength of agreement for the 10% overlap sample is lower, both authors will review all remaining abstracts. Any disagreements between the reviewers regarding suitability for inclusion will be rectified by joint decision and, if the dispute continues, a third reviewer (JD) will decide. Full-text screening will be performed for studies deemed eligible or of unclear eligibility, again by the same two independent reviewers and a third reviewer where disagreement regarding inclusion persists. ACM and LT will perform backward and forward citation searches to identify any other relevant literature. To establish agreement, this process will be conducted by both review authors (ACM and LT) for 10% of the included reference lists, and the remainder evenly distributed between the authors for independent screening. Further study inclusion will be determined using the previously described consensus approach. All article exclusions will be recorded and will be justified for any articles removed at the full-text stage.

### Data items and extraction process

Each paper selected for inclusion will be reviewed and data contained within the paper will be extracted and recorded using a data extraction form adapted from the Cochrane Review Group Data Extraction Form (online supplemental file 9). Both review authors (ACM and LT) will perform data extraction and any disagreements regarding this procedure will be discussed and settled by a third reviewer (JD) if required. Before starting, extraction consistency will be examined by piloting the form with 10% of eligible studies. Any disagreements arising from this process will be resolved through discussion between review authors (ACM and LT), and a third reviewer will decide on unresolved disagreements (JD). If needed, supplementary information will be requested directly from study authors to finalise inconsistencies and complete extraction.

For both qualitative and quantitative papers, extracted data will include—study characteristics (eg, aims, design, sample size and analysis), information source (eg, journal article or conference abstract), country and date of origin, title and publication status. Participant descriptive data, including age, gender and mental/physical health status will be recorded. Where provided, details regarding the actigraphic device will also be recorded, including device name, purpose (eg, sleep, obesity, ADHD symptom tracking), features (eg, material, design and functionality), body-worn location and mode of delivery (eg, on its own or part of a multi-faceted tool).

All text relating to relevant qualitative findings will be collected from the Results section of included papers, to record themes and subthemes pertaining to child-worn actigraph acceptability and engagement. Particular attention will be paid to any design or contextual factors that hinder or promote actigraphic device use in the target population. Where available, this will include direct participant quotes to afford greater data granularity at the data synthesis stage. Additionally, descriptions of the research methods used to assess acceptability (eg, structured questionnaire with space to report free text, structured or semi-structured interview or focus groups) and the information reporting source (eg, child, caregiver, or teacher) will be collected.

Where provided, actigraph monitoring protocol adherence data will be extracted—that is, the proportion of time in the given measurement period that the actigraphic device was actually worn for the duration specified in the study methods, according to self-report for example, diary entry or activity monitoring device data. We will also extract absolute numbers for the duration and intensity of prescribed and actual wear time, to investigate adherence according to protocol differences. Attrition rates and protocol deviations, that is, the percentage of consented participants who stop wearing the study device or wore the device in a different way than directed within the observed timeframe as well as any comments for this behaviour will be captured. Where studies also use questionnaires, scales or other methods to ascertain

acceptability/useability, the name and content of the method as well as participants' responses will be recorded. Information on factors which could influence participant device engagement will also be extracted. This could include but is not limited to, frequency of contact with the research team, explicit reminders to wear the device, for example, phone calls from the research team or automated text or email reminders and incentivisation. As this review focuses on the acceptability rather than the effectiveness of actigraphs, data relating to changes in behavioural outcomes following the use of an actigraphic device (eg, improved sleep or increased physical activity) will not be collected.

If a study is found which includes at least 50% of the children aged 5–11 years as well as a minority outside of this age group, we will aim to extract data related only to the age group of interest. If it is not possible to do so, this will be highlighted in the finalised report.

### Quality assessment

The relevant Critical Appraisal Skills Programme UK[41] checklists will be used to assess the quality of included articles according to study design. Quality assessment for each paper included in the review will be independently conducted by two review authors (ACM and LT), and any appraisal discrepancies will be resolved by a third reviewer (JD). Studies will not be excluded based on methodological quality, as the primary aim of this review is to identify literature about end users' experiences interacting with actigraphs. Rather, the appraisal will be used to comment on the general quality of available literature regarding the acceptability of body-worn actigraphic devices in children.

### Data synthesis and analysis

This review will use a systematic narrative synthesis approach.[42] The characteristics and key results for all included studies will first be presented in a table and elaborated on in the main body of the text. Summarised data will include study aims, the types of actigraphic devices, details about the participant group and specific constructs of acceptability that were assessed.

For qualitative studies, the main themes, data collection method and analysis method (eg, thematic, grounded theory or content analysis) will be presented. As outlined by Levitt,[43] to meta-analyse the qualitative studies, data relevant to the review question will be selected from the included studies to form 'meaning units', which will be compared with those from other studies to develop common themes and higher order categories. Depending on the number and type of papers identified by the search, it may be more appropriate to simply summarise the findings at this stage. However, to allow meaningful comparisons, where possible, the results will also be presented by subgroups, that is, according to device types or specific participant factors to better surmise any aesthetic requirements that are shared between or are unique to certain populations. Potential subgroups include, but will not be limited to, whether the children of interest are typically or atypically developing or have been diagnosed with a mental or physical health condition. Similarly, as the eligibility criteria span a period of considerable developmental change, the patterns of acceptability by age will be explored as well as gender and ethnicity.

Descriptive statistics relating to acceptability will be summarised for quantitative studies, including percentages, frequencies and measures of central tendency. If multiple studies are included that overlap sufficiently in methodology and outcome variables, random effects meta-analytical procedures will be used. For example, if possible, a pooled estimate of proportional wear time will be produced accompanied by a 95% CI using the Stata command Metaprop. The same procedure will be applied where two or more studies employ the same acceptability questionnaires or other outcome measures. For example, if compliance data are presented as a binary outcome, such as a frequency count of participants deviating from prescribed wear protocol or proportion failing to achieve a minimum wear time, the relative risk for that outcome will be calculated and meta-analytic methods for dichotomous outcomes will be considered, Statistical heterogeneity will be reported using the $I^2$ statistic. If 10 or more studies are identified for inclusion within a meta-analysis, techniques will be used to assess publication bias where appropriate, including Egger's regression test and funnel plots. Where possible, results stratified by actigraphic device type will be determined, and if the data permit, subgroup analyses will be reported according to the participant variables described for the qualitative subgroup analyses strategy.

### Patient and public involvement

No patients were involved in the conception or development of this project.

## DISCUSSION

This systematic review seeks to identify and summarise the existing literature relating to actigraphic device acceptability in young children and highlight particular features that make actigraphic products more or less palatable in childhood. To our knowledge, this is the first synthesis to focus on child and parental acceptance of actigraph use in free living settings for this age group, despite the potential impact of acceptability on wear time.[30 32]

Our intention to include studies which focus on both the subjective, experiential reports from children and caregivers as well as objective markers of wear time and attrition will allow a more comprehensive understanding of the role particular features within actigraphs play. For example, if qualitative reports consistently raise a feature as relevant, it may be possible to use a quantitative measure to corroborate this by comparing studies using devices with that feature to those without. Conversely, this approach may contradict the qualitative reports,

thus identifying areas for further exploration and future research.

We hope that, by including studies with a range of prescribed wear times as well as focusing primarily on community settings, our findings will have an impact on real-world research and clinical applications. For example, some features may promote compliance in the short term only, whereas another feature is associated with better prolonged wear. This knowledge, along with an understanding of the preferences of specific subgroups, would allow clinicians and researchers to select the optimal device for their needs and support designers to develop new products targeted to specific applications. This is timely given the rise of commercially available actigraphs purposefully developed for this age group.[36]

Furthermore, by collating and meta-analysing wear time and attrition data from several studies, we hope to set a benchmark, against which new devices may be compared. Such a benchmark could be helpful to device designers, particularly those wishing to evaluate prototypes against existing models in an iterative design process.

To strengthen this review, we have used a theoretical framework, the TFA to develop the search strategy as well as use broad search terms to maximise the chances of identifying papers relevant to the study question. A possible limitation is that the process of deducing themes across varied qualitative studies has been criticised owing to the potential interference of reviewer subjectivity on the study findings.[44] However, we have brought together a team of reviewers from multidisciplinary backgrounds to minimise any undue influence from any one reviewer's judgement. Our decision to exclude non-English written literature may limit the number of papers identified but is necessary due to resource constraints.

## ETHICS AND DISSEMINATION

This systematic review will use existing data and, therefore, does not require ethical approval. Findings from the review will be published in a peer-review journal and used to inform the development of a new actigraphic device.

**Contributors** ACM (corresponding author) and LT coconceived and developed the review protocol, defined the search strategy, drafted the initial, revised and finalised manuscripts. ACM and LT will perform data extraction and analysis and interpret the review findings. AW and SE provided critical revisions to the study design and manuscript. JD, ES-B and FM coconceived the review, supported the study design and amendments to the manuscript.

**Funding** This review is funded by the National Institute for Health Research (NIHR) Clinician Science Fellowship award (CS-2018-18-ST2-014). LT is funded by the NHIR as an Academic Clinical Fellow. AW is in receipt of a PhD studentship funded by the NIHR Biomedical Research Centre (BRC) at South London and Maudsley NHS Foundation Trust and Kings College London (NIHR-INF-0690). SE is funded by a Medical Research Council (MRC) Clinical Research Training Fellowship (MR/T001437/1) and previously received salary support from an MQ Data Science Award and from the Psychiatry Research Trust. JD is funded by the Clinician Science Fellowship award (CS-2018-18-ST2-014) and received additional funding from an MRC Clinical Research Training Fellowship (MR/L017105/1) and Psychiatry Research Trust Peggy Pollak Research Fellowship in Developmental Psychiatry. ESB is funded by the BRC, MRC (MR/T046864/1), Economic and Social Research Council (ESRC; ES/S004467/1), UK Research and Innovation's (UKRI) ESRC (ES/V016393/1) and NIHR Programme Grants for Applied Research (PGfAR; RP-PG-0618-20003).

**Competing interests** None declared.

**Patient and public involvement** Patients and/or the public were not involved in the design, or conduct, or reporting, or dissemination plans of this research.

**Patient consent for publication** Not applicable.

**Provenance and peer review** Not commissioned; externally peer reviewed.

**ORCID iDs**
Anna Charlotte Morris http://orcid.org/0000-0002-8691-9653
Alice Wickersham http://orcid.org/0000-0002-7402-7690
Sophie Epstein http://orcid.org/0000-0002-2118-908X
Johnny Downs http://orcid.org/0000-0002-8061-295X

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
