## [Reviewer comments · BMJ Open]

ARTICLE DETAILS

TITLE (PROVISIONAL)	Examining the acceptability of actigraphic devices in children using qualitative and quantitative approaches: protocol for a systematic review and meta-analysis
AUTHORS	Morris, Anna; Telesia, Laurence; Wickersham, Alice; Epstein, Sophie; Matcham, Faith; Sonuga-Barke, Edmund; Downs, Johnny

VERSION 1 – REVIEW

REVIEWER	Małoz, Piotr University of Rzeszów, Faculty of Physical Education
REVIEW RETURNED	18-Jun-2021

GENERAL COMMENTS	Thank you for the opportunity to review this valid and scientifically sound manuscript. Authors present detailed description of systematic review protocol they plan to implement in future project aimed to examine the acceptability of actigraphic devices in children using qualitative and quantitative approaches. Since this systematic review will use existing data therefore ethical soundness shouldn't be questioned in my opinion. The proposed protocol methodology is in my opinion proper and adequate to that the study will be conducted and analysed properly.
---

REVIEWER	Salvador-García, Celina Universidad Internacional de La Rioja
REVIEW RETURNED	24-Jul-2021

GENERAL COMMENTS	This protocol was interesting and easy to read. Without doubt, systematic review and meta-analysis proposed is relevant in order to understand users' perceptions on actigraphic devices that will inform the design of new devices. In addition, ideas are clearly presented and the text is easy to read most of the time. However, in my opinion, there are a few aspects that could be considered in order to enhance the quality of the paper. -I would suggest the authors to clearly present inclusion and exclusion criteria. In study design section they present these ideas, but, I think, a clear list presenting these criteria separately could clarify the text. In addition, this may be helpful when carrying out the process because each inclusion and exclusion criteria could be given a number and researchers could easily specify whether the texts fulfil or not these criteria. -Information sources are clearly presented and medicine-focused databases are a good choice for the topic under review. However, there may be other databases focused on education or social sciences in general, for example, in which authors could find
---

	articles concerning the subject under study. I would suggest them to include other database(s). -Concerning study selection process, I wonder why authors determined that each of the two investigators would screen 50% of the studies and only a 10% overlap. This could be justified. -In my opinion, data analysis regarding both qualitative and quantitative data could be a bit more specific. Other steps of the process are very precise. Nevertheless, data analysis does not seem so clearly depicted. For example, I wonder how authors will analyse the qualitative data gathered. -Authors state "As patient-centred design and acceptability literature is an emerging field we will extend inclusion to conference abstracts, commentaries, opinion pieces, editorials, letters, magazine articles, and technical or internal reports." Personally, I would reconsider this as well as not taking into account methodological quality as inclusion criteria (p.8). I understand the primary aim of their review is to identify literature pertaining to end users' experiences interacting with actigraphs. However, in my opinion, systematic reviews and meta-analyses should consider only robust and relevant sources for the information to be as pertinent as possible. -Regarding the list of terms used to carry out the search, authors could consider adding the word "opinion" for the last category. This is just a suggestion. -Authors could consider including the evaluation of the methodological quality of the systematic review using the 11-items checklist elaborated by AMSTAR, a measurement tool to assess the methodological quality of systematic reviews (Shea et al., 2007). This could help them increase the quality and empirical soundness of their systematic review and meta-analysis. -Discussion could be further developed. Although there are no results yet, I think a few more interesting ideas could be included in the current discussion section that will aid authors to construct their discussion once the protocol has been carried out and the results derived from the systematic review and meta-analysis have been presented.
--	---

VERSION 1 – AUTHOR RESPONSE

Reviewer 1

Comments to the authors:

- 1) *Thank you for the opportunity to review this valid and scientifically sound manuscript. Authors present detailed description of systematic review protocol they plan to implement in future project aimed to examine the acceptability of actigraphic devices in children using qualitative and quantitative approaches. Since this systematic review will use existing data therefore ethical soundness shouldn't be questioned in my opinion. The proposed protocol methodology is in my opinion proper and adequate to that the study will be conducted and analysed properly.*

Authors' response:

We appreciate you taking the time to review our manuscript and thank you for your positive feedback.

Reviewer 2

Comments to the authors:

- 1) *I would suggest the authors to clearly present inclusion and exclusion criteria. In study design section they present these ideas, but, I think, a clear list presenting these criteria separately could clarify the text. In addition, this may be helpful when carrying out the process because each inclusion and exclusion criteria could be given a number and researchers could easily specify whether the texts fulfil or not these criteria.*

Authors' response:

To clarify the inclusion and exclusion criteria beyond what is in the text, we have amended the manuscript to include the following table:

	Inclusion criteria	Exclusion criteria
Publication status	 - Studies written in English - Published or unpublished studies 	 - Studies written in other languages - Commentaries, opinion pieces, editorials, magazine articles and technical or internal reports
Study design	 - Qualitative or quantitative studies 	 - Reviews or meta-analyses
Device	 - Studies in which a wearable activity tracker to measure motor activity was worn by a child within the target age group 	 - Studies which do not employ any wearable devices - Studies using wearable devices which measure parameters other than motor activity (e.g., balance, number of steps, proportion of time sitting etc.) - Studies in which wearable activity trackers were discussed, but not worn
Age	 - Studies where all participants are aged between 5-11 years, as indicated by age range, or reported school year - Studies reporting a measure of central tendency (e.g., median, mean) which falls within the range of 5-11 years - Studies which present data in a way that allows extraction for a subgroup aged between 5-11 years - Studies that satisfy the above age criteria for the child wearing the device, but feedback is 	 - Studies where all participants are outside the target age of 5-11 years - Studies where it is not possible to extract data specific for participants aged between 5-11 years

		provided by a caregiver or other relevant stakeholder	
Setting	-	Studies with protocols which involves the wear of an activity tracker in a community setting, regardless of whether the device is also worn in a non-community setting	- Studies which take place entirely in a non-community setting (e.g., laboratory, university or hospital setting)
Acceptability	-	Studies which present a qualitative or quantitative measure of acceptability - Measures can include focus group or interview feedback, proportions of participants declining to wear the device, or proportion and extent of deviation from the study's wear protocol	- Studies which do not contain any direct or indirect measure of acceptability

2) *Information sources are clearly presented and medicine-focused databases are a good choice for the topic under review. However, there may be other databases focused on education or social sciences in general, for example, in which authors could find articles concerning the subject under study. I would suggest them to include other database(s).*

Authors' response:

In accordance with this suggestion, the search strategy has been amended to include additional relevant databases. The text now reads:

“The electronic databases Embase, MEDLINE, PsychInfo and Social Policy and Practice through the OVID interface; and Education Resources Information Center (ERIC), British Education Index and CINAHL through the EBSCO interface will be searched using a combination of free text and thesaurus terms.”

3) *Concerning study selection process, I wonder why authors determined that each of the two investigators would screen 50% of the studies and only a 10% overlap. This could be justified.*

Authors' response:

It was determined that, if the 10% overlap returned a sufficiently high level of agreement, both investigators reviewing 100% of the abstracts would not be necessary. We feel this is consistent with the AMSTAR tool, referenced below. The text has been updated to clarify this point, and now reads:

“Two reviewers (AM and LT) will each independently screen a 10% sample of the titles and abstracts generated by the search strategy. Interrater reliability for this sample will be determined using Cohen's kappa.³⁹ If the interrater reliability is considered “substantial” or

“almost perfect”, based on a kappa statistic greater than 0.60 (Landis & Koch, 1977), each review author will independently review 50% of the remaining abstracts, with the option to jointly discuss unclear or otherwise challenging abstracts. If the strength of agreement for the 10% overlap sample is lower, both authors will review all remaining abstracts. Any disagreements between the reviewers regarding suitability for inclusion will be rectified by joint decision and, if the dispute continues, a third reviewer (JD) will decide.”

- 4) *In my opinion, data analysis regarding both qualitative and quantitative data could be a bit more specific. Other steps of the process are very precise. Nevertheless, data analysis does not seem so clearly depicted. For example, I wonder how authors will analyse the qualitative data gathered.*

Authors' response:

We have chosen to allow some flexibility with regards to how acceptability data may be presented within the included studies. To a large extent, the most suitable data analysis plan will depend on this presentation and the number of studies included from the search. However, we do anticipate several possible methods of presentation. For experiential data, we expect this to be mostly qualitative (via interview or focus groups) with some studies using questionnaires. For compliance data as an implied measure of acceptability, data may be presented in a variety of ways, including frequency counts of patients deviating from prescribed wear protocol, frequency counts with extent of deviation from the wear protocol (e.g., number wearing the device for specific time periods, or number failing to meet a threshold), or average wear time for the whole sample or a subsample. We have updated the text to reflect this, and it now reads:

“This review will utilise a systematic narrative synthesis approach. ⁴¹ The characteristics and key results for all included studies will first be presented in a table and elaborated on in the main body of text. Summarised data will include, study aims, the types of actigraphic devices, details about the participant group and specific constructs of acceptability that were assessed.

For qualitative studies, the main themes, data collection method and analysis method (e.g. thematic, grounded theory or content analysis) will be presented. As outlined by Levitt, 2018, to meta-analyse the qualitative studies, data relevant to the review question will be selected from the included studies to form “meaning units” which will be compared with those from other studies to develop common themes and higher order categories. Depending on the number and type of papers identified by the search, it may be more appropriate to simply summarise the findings at this stage. However, to allow meaningful comparisons, where possible the results will also be presented by subgroups, that is, according to device types or specific participant factors to better surmise any aesthetic requirements that are shared between or are unique to certain populations.”

and

“Descriptive statistics relating to acceptability will be summarised for quantitative studies, including percentages, frequencies, and measures of central tendency. If multiple studies are included which overlap sufficiently in methodology and outcome variables, random effects meta-analytical procedures will be used. For example, if possible, a pooled estimate of proportional wear time will be produced accompanied by a 95% confidence interval (CI) using the Stata command Metaprop. The same procedure will be applied where two or more studies employ the same acceptability questionnaires or other outcome measure. For example, if compliance data is presented as a binary outcome, such as a frequency count of participants deviating from prescribed wear protocol or proportion failing to achieve a minimum wear-time, the relative risk for that outcome will be calculated and meta-analytic methods for

dichotomous outcomes will be considered, Statistical heterogeneity will be reported using the I^2 statistic. If ten or more studies are identified for inclusion within a meta-analysis, techniques will be used to assess publication bias where appropriate, including Egger's regression test and funnel plots. Where possible, results stratified by actigraphic device type will be determined, and if the data permits, subgroup analyses will be reported according to the participant variables described for the qualitative subgroup analyses strategy."

- 5) *Authors state "As patient-centred design and acceptability literature is an emerging field we will extend inclusion to conference abstracts, commentaries, opinion pieces, editorials, letters, magazine articles, and technical or internal reports." Personally, I would reconsider this as well as not taking into account methodological quality as inclusion criteria (p.8). I understand the primary aim of their review is to identify literature pertaining to end users' experiences interacting with actigraphs. However, in my opinion, systematic reviews and meta-analyses should consider only robust and relevant sources for the information to be as pertinent as possible.*

Authors' response:

a. In response to the first point, we have made amendments to the text. We feel that conference abstracts and letters have the potential to be robust sources of data for this project, provided we are consistent with the inclusion and exclusion criteria. Other sources are less likely to provide high quality data. The amended text now reads:

"Qualitative studies, specifically co-design, focus groups and interview methodologies will be included as well as quantitative studies including feasibility, pilot, observational cohort, or intervention studies. Reviews, meta-analyses, and protocols will be excluded, as will commentaries, opinion pieces, editorials, magazine articles, and technical or internal reports. As patient-centred design and acceptability literature is an emerging field we will extend inclusion to published conference abstracts if they provide sufficient information to demonstrate that they meet the inclusion criteria. Likewise, letters that describe the methods and results of otherwise unpublished research may also be considered for inclusion. No date restrictions will be applied to this search. Studies can be published or unpublished but must be written in English."

b. Regarding the methodological quality as a potential inclusion criterion, we feel it would be most beneficial to present all the data available, and then comment on its quality within the review. Additionally, we intend to, where possible, infer findings pertaining to acceptability from papers where acceptability is not necessarily the primary focus of the study. As such, a paper with otherwise poor methodology may provide data of sufficient quality for our purposes.

- 6) *Regarding the list of terms used to carry out the search, authors could consider adding the word "opinion" for the last category. This is just a suggestion.*

Authors' response:

The search terms have been amended to include "opinion" within the last category. The updated search terms, also available in the supplementary documents, now read:

1. child*.ab,ti
2. exp child/
3. primary school.ab,ti.

4. youth*.ab,ti.
5. exp juvenile/
6. kindergar*en.ab,ti.
7. (school adj2 (children or kid* or pupil*)).ab,ti.
8. young people*.ab,ti
9. 1or2or3or4or5or6or7or8
10. (actigraph* or actimet* or actograp* or actomet* or acceleromet*).ab,ti.
11. "motor activity".ab,ti.
12. Fitbit.ab,ti.
13. ((electronic or remote or wearable or fitness or activity) adj2 (track* or monitor* or wearable* or device* or technolo*)).ab,ti.
14. step count.ab,ti
15. 10 or 11 or 12 or 13 or 14
16. acceptability.ab,ti.
17. experience*.ab,ti.
18. perception*.ab,ti.
19. feasibility.ab,ti.
20. feedback.ab,ti.
21. design*.ab,ti.
22. usability.ab,ti.
23. practicability.ab,ti.
24. willingness.ab,ti.
25. usefulness.ab,ti.
26. engagement.ab,ti.
27. opinion.ab,ti
28. 16 or 17 or 18 or 19 or 20 or 21 or 22 or 23 or 24 or 25 or 26 or 27
29. 9 and 15 and 28

7) *Authors could consider including the evaluation of the methodological quality of the systematic review using the 11-items checklist elaborated by AMSTAR, a measurement tool to assess the methodological quality of systematic reviews (Shea et al., 2007). This could help them increase the quality and empirical soundness of their systematic review and meta-analysis.*

Authors' response:

We appreciate you drawing our attention to this useful resource. Our understanding is that it is primarily aimed at reviews which investigate the effect of a healthcare intervention, but it is also helpful to consider the items that are pertinent to the present project. We have reviewed those relevant to the design phase of systematic reviews, and feel our current protocol to be a robust one. This tool will also have value in the next stages, as we continue with this project.

7) *Discussion could be further developed. Although there are no results yet, I think a few more interesting ideas could be included in the current discussion section that will aid authors to construct their discussion once the protocol has been carried out and the results derived from the systematic review and meta-analysis have been presented.*

Authors' response:

We have built upon the existing text, which now includes the following:

“Our intention to include studies which focus on both the subjective, experiential reports from children and caregivers, as well as objective markers of wear time and attrition will allow a more comprehensive understanding of the role particular features within actigraphs play. For example, if qualitative reports consistently raise a feature as relevant, it may be possible to use a quantitative measure to corroborate this by comparing studies using devices with that feature to those without. Conversely, this approach may contradict the qualitative reports, thus identifying areas for further exploration and future research.

We hope that, by including studies with a range of prescribed wear times as well as focusing primarily on community settings, our findings will have an impact on real world research and clinical applications. For example, some features may promote compliance in the short term only, whereas another feature is associated with better prolonged wear. This knowledge, along with understanding of the preferences of specific subgroups, would allow clinicians and researchers to select the optimal device for their needs, and support designers to develop new products targeted to specific applications. This is timely given the rise of commercially available actigraphs purposefully developed for this age group.³⁶

Furthermore, by collating and meta-analysing wear time and attrition data from several studies we hope to set a benchmark, against which new devices may be compared. Such a benchmark could be helpful to device designers, particularly those wishing to evaluate prototypes against existing products in an iterative design process.”

VERSION 2 – REVIEW

REVIEWER	Salvador-García, Celina Universidad Internacional de La Rioja
REVIEW RETURNED	10-Dec-2022
GENERAL COMMENTS	Thank you for letting me review this manuscript again. As I mentioned in the previous round of review, the protocol presented is relevant in order to understand users’ perceptions on actigraphic devices that will inform the design of new devices. In addition, the authors have improved the manuscript and they have addressed many of the comments I posed. There is just a minor aspect that the authors could consider. In the new version of the paper, some new databases have been incorporated in the main text. In this sense, these newly added databases could be added to those that already appeared in abstract.

VERSION 2 – AUTHOR RESPONSE

Reviewer: 1

Thank you for letting me review this manuscript again. As I mentioned in the previous round of review, the protocol presented is relevant in order to understand users’ perceptions on actigraphic devices that will inform the design of new devices. In addition, the authors have improved the manuscript and they have addressed many of the comments I posed. There is just a minor aspect that the authors could consider. In the new version of the paper, some new databases have been incorporated in the main text. In this sense, these newly added databases could be added to those that already appeared in abstract.

Authors response:

Many thanks for highlighting this discrepancy, we have updated the abstract to include all relevant databases as follows: Embase, MEDLINE, PsychInfo and Social Policy and Practice through the OVID interface; and Education Resources Information Center (ERIC), British Education Index and CINAHL through the EBSCO interface.